Social buffering and contact transmission: network connections have beneficial and detrimental effects on Shigella infection risk among captive rhesus macaques

Balasubramaniam Krishna krishnanatarajan@ucdavis.edu 1
Beisner Brianne 1 2
Vandeleest Jessica 1 2
Atwill Edward 1
McCowan Brenda 1 2
1 Department of Population Health & Reproduction, School of Veterinary Medicine, University of California , Davis , CA , United States
2 Brain, Mind & Behavior, California National Primate Research Center, University of California , Davis , CA , United States
McMullan Rachel
Electronic publication date: 2016 Oct 27
Publication date: 2016
Volume: 4
Electronic Location ID: e2630
Received 2016 Jun 15; Accepted 2016 Sep 29
Copyright: ©2016 Balasubramaniam et al.
Copyright year: 2016
Copyright holder: Balasubramaniam et al.
License: This is an open access article distributed under the terms of the Creative Commons Attribution License, which permits unrestricted use, distribution, reproduction and adaptation in any medium and for any purpose provided that it is properly attributed. For attribution, the original author(s), title, publication source (PeerJ) and either DOI or URL of the article must be cited.
License URL: https://creativecommons.org/licenses/by/4.0/

Keywords: Social buffering, Nonhuman primate, Social networks, Contact-mediated transmission, Infectious disease risk

Funding: NICHD of the National Institutes of Health R01HD068335 Research reported in this publication was supported by NICHD of the National Institutes of Health under award number R01HD068335. The content is solely the responsibility of the authors and does not necessarily represent the official views of the National Institutes of Health. The funders had no role in study design, data collection and analysis, decision to publish, or preparation of the manuscript.

==============================
In social animals, group living may impact the risk of infectious disease acquisition in two ways. On the one hand, social connectedness puts individuals at greater risk or susceptibility for acquiring enteric pathogens via contact-mediated transmission. Yet conversely, in strongly bonded societies like humans and some nonhuman primates, having close connections and strong social ties of support can also socially buffer individuals against susceptibility or transmissibility of infectious agents. Using social network analyses, we assessed the potentially competing roles of contact-mediated transmission and social buffering on the risk of infection from an enteric bacterial pathogen (Shigella flexneri) among captive groups of rhesus macaques (Macaca mulatta). Our results indicate that, within two macaque groups, individuals possessing more direct and especially indirect connections in their grooming and huddling social networks were less susceptible to infection. These results are in sharp contrast to several previous studies that indicate that increased (direct) contact-mediated transmission facilitates infectious disease transmission, including our own findings in a third macaque group in which individuals central in their huddling network and/or which initiated more fights were more likely to be infected. In summary, our findings reveal that an individual’s social connections may increase or decrease its chances of acquiring infectious agents. They extend the applicability of the social buffering hypothesis, beyond just stress and immune-function-related health benefits, to the additional health outcome of infectious disease resistance. Finally, we speculate that the circumstances under which social buffering versus contact-mediated transmission may occur could depend on multiple factors, such as living condition, pathogen-specific transmission routes, and/or an overall social context such as a group’s social stability.

Introduction

In humans and other animals, the strength and diversity of social relationships strongly influence the risk of acquiring infectious diseases (Alexander, 1974; Drewe & Perkins, 2015; McCowan et al., 2016; Nunn, 2012). In addition to life-history traits of the host, biology of the pathogen, and the degree of contact with contaminated environmental sources (Drewe & Perkins, 2015; Kappeler, Cremer & Nunn, 2015), social connections with group conspecifics may also influence the risk of acquiring and/or transmitting a pathogen (Drewe & Perkins, 2015). This may occur in two ways. On the one hand, increased connections may lead to greater chances of infection from pathogens via contact-mediated transmission (Drewe & Perkins, 2015), making infectious disease acquisition a major cost of social living (Alexander, 1974; Freeland, 1976; Loehle, 1995; MacIntosh et al., 2012). Yet social connections may also mitigate the impact of stressors or immunosuppressive effects of stress, thereby socially buffering an individual to decrease their susceptibility to infection (Hennessy, Kaiser & Sachser, 2009; Kikusui, Winslow & Mori, 2006; McCowan et al., 2016; Sapolsky, 2005; Sapolsky, Romero & Munck, 2000; Segerstrom & Miller, 2004; Young et al., 2014). To better understand the impact of social life on disease risk, it is necessary to characterize the potentially competing impacts of both greater contact-mediated transmission and social buffering on susceptibility to, and transmission of pathogens.

Traditional models of disease acquisition and transmission in social systems assume that individuals interact randomly (Anderson & May, 1992). In reality, individuals differ in how they navigate their social environment, which reflects differences in their social strategies to maximize fitness (Alexander, 1974; Bansal, Grenfell & Meyers, 2007). Such heterogeneity can be modeled using social network analysis (Krause, Croft & James, 2007; McCowan et al., 2008). Most recent work implementing social network analysis to investigate infectious agent epidemiology in animal systems has focused on contact-mediated infection and transmission (reviewed in Drewe & Perkins, 2015). Broadly, this phenomenon predicts that greater social connectedness, via more frequent contact rates with infected individuals within one’s social network, increases one’s risk of being infected by socially-transmitted pathogens (Drewe & Perkins, 2015). Specifically, being very central, and/or showing increased rates of contact-associated behaviors (e.g., social or allogrooming) within a social network has been shown to impact a variety of infectious-disease associated health outcomes, such as increasing endoparasite load, prevalence of a specific pathogen, and pathogenic diversity of individuals (Nunn, 2012). Indeed, contact-mediated transmission of pathogens has been shown to occur among a variety of animal social groups, such as group-living lizards (Godfrey et al., 2009), Tasmanian devils (Hamede et al., 2009), Belding’s ground-squirrels (VanderWaal et al., 2013), and nonhuman primates (MacIntosh et al., 2012; Rimbach et al., 2015). Furthermore, epidemiologists speculate that highly central individuals may also act as superspreaders of pathogens because they can also transmit an infection to their neighbors (Drewe & Perkins, 2015; Lloyd-Smith et al., 2005). Yet an individual’s susceptibility versus resistance to infection may depend not just on its direct connections but also on its neighbors’ connections (Farine & Whitehead, 2015). To date, the impact of such indirect or secondary connections on infection risk has received relatively less attention in epidemiological studies (but see MacIntosh et al., 2012).

Even less explored is the possibility that having more direct or indirect connections in a network may, via social buffering, protect individuals from social and environmental stressors (Kaplan et al., 1991; Young et al., 2014), and decrease rather than increase their susceptibilities to infection. In societies where individuals maintain strong social bonds (e.g., humans, some nonhuman primates like baboons (Papio sp.) and macaques (Macaca sp.)), social buffering is well-documented—having more social ties can mitigate stress to generate positive health outcomes (Cobb, 1976; Young et al., 2014) and indeed, enhance longevity and survival (Archie et al., 2014; Silk et al., 2010). In humans for example, having more connections who provide social support during times of conflict alleviates stress-levels (Janowski et al., 2012) and decreases susceptibility to non-communicable diseases such as cancer and cardiovascular disease (Uchino, 2004; Uchino, 2009). In nonhuman primates, grooming (an affiliative social interaction) may be exchanged among group members for access to social support (Seyfarth, 1977), and is also known to lower circulating levels of glucocorticoids by reducing activation of the hypothalamic-pituitary-adrenal (HPA) axis (Young et al., 2014). Socio-positive interactions appear to be key in alleviating stress, which both enhances immune function (Sapolsky, Romero & Munck, 2000) and decreases susceptibility to infectious agents (Cohen, Janicki-Deverts & Miller, 2007; Segerstrom & Miller, 2004). Yet to our knowledge, no work has investigated the effects of social buffering on health using network-based analytical approaches, or in the context of infectious diseases. To better understand how social connections impact disease-related health outcomes, it is imperative to assess the effects of contact-mediated transmission and social buffering, particularly indirect social connections through which group members may elicit social support, on the risk of acquiring infectious agents.

Here we use a nonhuman primate model, the rhesus macaque (Macaca mulatta), to test whether the risk of infection from a bacterial pathogen (Shigella flexneri) is higher in socially well-connected individuals or lower in individuals buffered by having access to social support. Enteric bacterial pathogens, such as Shigella, are ubiquitous, well documented, and among the most communicable pathogens in humans (DuPont, 2000). Although infected individuals can occasionally remain asymptomatic, doses as low as between 20 and 200 organisms may be sufficient to cause symptoms of Shigellosis, ranging from acute to severe diarrhea, fever, and even death (DuPont, 2000). Shigella transmission in humans occurs via the fecal-oral route, or person-to-person contact, as well as consumption of contaminated food and water sources (Anderson & May, 1992; Scallan et al., 2011). Among nonhuman primates, Shigella (and indeed other enteric bacteria) are surprisingly understudied in comparison with other pathogens (http://www.mammalparasites.org) (Nunn & Altizer, 2006). The few studies to date report low prevalence in some free-living populations that live in close proximity to humans (e.g., rhesus macaques: Beisner et al., 2016; Savannah baboons: Drewe et al., 2012; Harper et al., 2012). Although the dosage levels of Shigella flexneri that may generate virulent effects among primates is unclear, the occurrence of acute enteritis in rhesus macaques housed in captivity has been linked to infection from this pathogen (Lee, Kim & Park, 2011). There have also been two documented outbreaks of Shigellosis among semi-free ranging rhesus macaques at Cayo Santiago, Puerto Rico, resulting in enteritis, abortions, and increased mortality rates among pregnant monkeys (Kessler & Rawlins, 2016). Such documentations of virulent infection among primates, further to its well-established social- and/or environmental-contact mediated transmission routes among humans, make Shigella an ideal pathogen for investigating the effects of social network connectedness on susceptibility versus resistance to infection risk.

Rhesus macaques are biologically, socially and cognitively analogous to human societies (Cobb, 1976; Suomi, 2011). They live in large (∼20–150 individuals), multi-male-multi-female social groups (Thierry, 2007). Individuals maintain and reinforce their social relationships using a variety of behaviors, such as aggression, grooming, and social huddling (Lindburg, 1971; Sade, 1972). These behaviors are heterogeneously distributed in accordance with sex, kinship, age-sex category, and dominance ranks of individuals (Thierry, 2007). For example, social bonds established via grooming tend to be strongest among closely related females, and/or females with adjacent dominance rank which is also inherited in accordance with kinship (e.g., Berman & Thierry, 2010; Chapais, 2006; Sade, 1972). From a health perspective, these behaviors typically bring individuals into physical contact and can also influence levels of social tension or stress. For example, grooming has well-documented fitness, particularly stress-relieving benefits in nonhuman primates (summarized in Henzi & Barrett, 1999) as well as in other animal taxa (e.g., social insects: Moore et al., 1995 horses: Kimura, 1998). Similarly, although the functional significance of huddling among macaques is yet to be documented, this behavior may be considered to be physiologically (involving relaxed body-to-body contact) analogous to hugging among humans, which has been shown to socially buffer individuals from infection by the common-cold virus (Cohen et al., 2015). Conversely, grooming can also serve as a contact-route of infection (Japanese macaques (M. fuscata: MacIntosh et al., 2012); other primates such as brown spider monkeys (Ateles hybridus: Rimbach et al., 2015)). Aggressive interactions have the potential to enhance susceptibility to infection by bringing aggressors and recipients into physical contact (Drewe, 2010), or by elevating stress-levels in recipients of aggression (e.g.,  Muller & Wrangham, 2004). Among Japanese macaques (MacIntosh et al., 2012) revealed that dominance rank, an outcome of dyadic aggressive interactions, positively affected parasite diversity but not fecal glucocorticoids, indicating a contact-rather than a stress-mediated infection-route. In primate species like rhesus macaques that are characterized by despotic social relationships, the effect of dominance rank on infection risk may be somewhat redundant to examining those of social network connectedness, since social rank directly effects such connectedness and centrality (e.g.,  MacIntosh et al., 2012; Sueur et al., 2011b). Yet other than MacIntosh et al. (2012), we are unaware of any work that has simultaneously examined the effect(s) of greater network connectedness (which may occur via dominance rank) to specifically evaluate the contrasting roles of contact-mediated transmission and social buffering on infectious disease risk.

We first establish (1) the prevalence of Shigella flexneri within each of three groups of captive rhesus macaques. For each of these groups, we next reconstruct each of three types of social networks based on grooming, huddling, and aggressive interactions respectively. Upon doing so, we investigated whether (2) greater centrality and/or social connectedness through both direct and (where relevant) indirect or secondary connections in these networks mitigated macaques’ risk of infection via social buffering, versus enhanced this risk through contact-mediated transmission. Specifically, if connections socially buffer individuals against Shigella infection risk, we predicted that this bacterial pathogen will be the least prevalent among individuals with the most direct and/or indirect social connections. On the other hand, if possessing connections enhance infection risk via contact-mediated transmission, we predicted that infection will be most prevalent among individuals with the highest number or diversity of direct connections. Finally, we also ascertained whether (3) attributes of individuals other than their network positions, but which may influence their positions (particularly sex, dominance rank, and the certainty of their dominance relationships (Fujii et al., 2014, unpublished data; see ‘Methods’)), also influenced their risk of infection. Specifically, we asked whether Shigella infection was least likely among high-ranking females who typically form the strongest social bonds. Further, given that increased uncertainty in dominance relationships could be more stressful (Vandeleest et al., 2016), we asked whether the effect of possessing increased connections on socially buffering individuals from infection risk was more clearly discernible among individuals who also had more uncertain compared to certain dominance relationships.

Materials and Methods

Study location and subjects

The study was conducted at the California National Primate Research Center (CNPRC) and the School of Veterinary Medicine (SVM), University of California at Davis. The subjects were 299 adult rhesus macaques (90 males, 209 females) between the ages of 3 and 29 (mean = 7.7 years), distributed across three social groups (101 in Group I, 96 in Group II, and 102 in Group III). The groups were housed in 0.2 ha outdoor enclosures containing multiple A-frame structures, suspended barrels, swings, and perches; they were free to engage in social interaction. Animals were fed a standard diet of monkey chow twice per day at approximately 0700 h and between 1,430 and 1,530 h. Fresh fruit or vegetables were provided one time per week and seed mixture provided daily. Water was available ad libitum. The outdoor housing facilities are exposed to a minimum level of disturbance, and may be considered semi-naturalistic. The protocols used for this research were approved by the UC Davis Institutional Animal Care and Use Committee (IACUC), and were in accordance with the legal requirements of the jurisdictions in which the research was conducted.

Behavioral data collection

Behavioral data were collected on each group for 6 weeks, two groups in the spring (Group I: March–April 2013; Group III: March–April 2014) and one in the fall (Group II: September–October 2014). For each group, three observers collected data for 6 h on 4 days per week from 0900–1200 h and 1300–1600 h, using (i) an Event Sampling design for aggressive interactions, and (ii) Scan Sampling for affiliative grooming and huddling interactions (Altmann, 1974). These have been shown to optimize reliable collection of data in large groups of subjects (McCowan et al., 2011), both improving statistical power and circumventing nonindependence issues that may affect the computation of reliable social network metrics (Farine & Whitehead, 2015). For each ‘event’, the identities of the initiator and recipient macaque, and their specific behavior(s) were recorded. Grooming was defined as an animal cleaning or manipulating the fur of another individual. Huddling was defined as the occurrence of all forms of body-contact, including (but not restricted to) ventral contact or an embrace between two individuals that did not involve a social behavioral interaction (e.g., grooming, contact aggression). Aggression was categorized according to severity and included threat (open mouth stare, brow flash, ear flap), mild aggression (threat and follow, lunge, push, slap, chase < 6 m), moderate aggression (grapple, wrestle, chase > 6 m), and intense aggression (pin or bite). We recorded events of polyadic aggression as a series of dyadic interactions. Data on all dyadic and polyadic interactions were used to calculate the network metrics of degree and strength. Submission categories included freeze/turn away, move away, run away < 6 m, run away > 6 m, and crouch. For computing dominance rank and certainty (see below), we used only data on dyadic aggressive and submissive interactions.

Social network and dominance metrics

For each of the three groups, we constructed three types of social networks: (1) groom, (2) huddle, and (3) aggression. From undirected grooming and huddling networks, we calculated the number of direct connections (or degree centrality), the total number of shortest pathways between other pairs of animals that pass through an individual (or betweenness centrality), and a metric of social capital that takes both direct and indirect connections of individuals into consideration (eigenvector centrality) (Farine & Whitehead, 2015; Makagon, McCowan & Mench, 2012). For directed networks of grooming and aggression, we also calculated the number of connections given and received (in- and out-degree respectively), and the strengths of these connections weighted by the frequencies of interactions (in- and out-strength respectively). These were computed using the Statnet and Sna (Handcock et al., 2006) packages in R. Given its estimation of an individual’s direct connections, degree centrality has been extensively used in epidemiological studies to date that have focused on contact-mediated pathogen transmission (Drewe, 2010; Drewe & Perkins, 2015; Rimbach et al., 2015). On the other hand, in- and out-degree and strength in grooming may also be indicative of reduced infection risk via social buffering, given that grooming interactions have well-documented benefits of lowering the physiological stress levels among both givers and receivers (Henzi & Barrett, 1999; Smutt et al., 2007; Young et al., 2014). Although betweenness centrality has been less commonly used in epidemiological studies, it has been proposed as being a key metric in predicting the potential for the flow of infectious agents through the generally more densely connected social networks (Drewe & Perkins, 2015; Farine & Whitehead, 2015; Newman, 2005; VanderWaal et al., 2014). Finally, as an indicator of access to social support via indexing secondary connections, eigenvector centrality may be particularly key in determining whether the possession of such extended support circles beneficially inhibits (via buffering), versus detrimentally exposes an individual to infection risk (via contact-transmission: e.g.,  MacIntosh et al., 2012).

We implemented a recently developed network algorithm, Percolation-and-Conductance (or Perc: Fujii et al., 2014, unpublished data), to compute dominance ranks and certainties from aggression networks constructed from dyadic interactions only. This method combines information from direct dominance interactions with information from multiple indirect dominance pathways (via common third parties) to quantify dyadic dominance relationships (Fujii et al., 2014, unpublished data). This algorithm identifies all potential flow pathways in the network, weighting the contribution of each path to the imputed matrix by its likelihood of being traversed by the random walk. The method yields two outputs: a matrix of dyadic dominance certainty values (range: 0–1) and the lowest-cost linear rank order. The former represent the cumulative information from all network pathways between each pair of animals. A dominance certainty value of 1 reflects the highest possible certainty that the row animal outranks the column animal (and 0 reflects the highest possible certainty that the column animal outranks the row animal) whereas 0.5 means the dominance relationship is perfectly ambiguous. To compute the average dominance certainty of each individual, we transformed all dyadic dominance certainty values between 0.5 and 1, and calculated the row-wise average for each animal. We also transformed ordinal dominance ranks for each group into the proportion of animals outranked within their respective groups (i.e., 0 is the lowest ranked animal and 1 is the highest ranked animal). This measure of ‘percentile ranks’ was used in the place of ordinal ranks in the analyses.

Pathogen characterization

Prior to fecal collection, animals were immobilized (10 mg/kg of ketamine) and given standard physical examinations by veterinary staff (e.g., checked for injuries, weighed). We collected two fresh fecal swabs from every macaque at the end of the behavioral observation period following previously published methods (Good, May & Kawatomari, 1969). Briefly, a sterile cotton-tip swab was inserted into the rectum of each individual, rotated gently to collect fecal material, and immediately immersed into a 15 ml test-tube (labeled with the animal ID) containing sterile Tryptic Soy Broth (TSB; BD, Franklin Lakes, NJ, USA); a duplicate sample was taken using another sterile swab and placed into a second TSB tube. The samples were incubated within 4 h of collection; the tubes were incubated with orbital rotation of 100 rpm at (1) 25 °C for 2 h, (2) 42 °C for 8 h, and (3) held static at 6 °C overnight. Shigella was isolated from the TSB enrichment; ∼10 µL enrichment was struck for isolation onto MacConkey agar plates (BD, Franklin Lakes, NJ, USA), and incubated at 37 °C for 18–24 h. Suspect colonies were further isolated and characterized on Xylose Lysine Deoxycholate agar plates (XLD; BD, Franklin Lakes, NJ, USA). Two isolated colonies per sample were biochemically confirmed using (i) Triple Sugar Iron (TSI) (Remel, Lenexa, KS), (ii) Citrate (Remel, Lenexa, KS, USA), (iii) Urea (BD, Franklin Lakes, NJ, USA), and (iv) the Methyl Red –Voges-Proskauer (MR-VP) test (BD, Franklin Lakes). Individuals which tested positive from at least one of the two swabs were categorized as infected, with those that tested negative from both swabs deemed uninfected.

Data analysis

The low prevalence of bacterial infection among macaques (e.g.,  Beisner et al., 2016; Drewe et al., 2012; see ‘Results’) has the potential to impact the statistical power of our analyses. We therefore conducted some diagnostic assessments to determine whether there was sufficient homogeneity of the data across groups to warrant combining them in a single model-set. Diagnostic plots (Fig. 1) showed that while the patterns of relationships between Shigella infection and network metrics appeared more similar for Groups I and II, those for Group III were starkly different. These are further supported by scatter-plots that showed similar infection patterns (concentrated among individuals moderately or least central in their grooming and huddling networks) in Groups I and II, but a different pattern of infection being widely distributed across individuals of varying centralities in Group III (Fig. S1). Finally, Groups I and II were highly similar, and different from Group III, in the age structure of individuals, time of formation, and the number of matrilines, any or all of which may impact heterogeneity in social network dynamics (Table S1; see ‘Discussion’). For these reasons, we chose to combine Groups I and II into a single, homogeneous population for our analyses, and analyzed Group III as a second, separate dataset.

Figure 1 Diagnostic box-plots of between-group similarities and differences in relationships between network metrics and Shigella infection.

Each plot shows the mean and distribution of a network metric (Y axis) plotted for infected and uninfected individuals in each group (X Axis). Red box-plots represent comparisons for Groups I and II, and Yellow comparisons for Group III.

To investigate our aims, we implemented an Information-Theoretical approach to construct generalized linear mixed-effects models (Burnham, Anderson & Huyvaert, 2011; Grueber et al., 2011; Whittingham et al., 2006). For each of the two datasets, we constructed 48 models using the lme4 package in R (Bates et al., 2016). We used a binomial distribution with a Logit link function to examine the effect of social network metrics, attributes, and their potential interactions on Shigella infection across individual macaques McCullagh & Nelder, 1989). Given the potential collinearity or non-independence among centrality network metrics (Farine & Whitehead, 2015; Krause, Croft & James, 2007; MacIntosh et al., 2012; Sueur et al., 2011a), we refrained from using the automated ‘Dredge’ function that provides a full set of sub-models for a set of predictors (Grueber et al., 2011). Instead, our complete model-set (48 models for each dataset: Tables S2 and S3) was composed of individually-constructed models, specifically a ‘null’ model (1 model), a model each for the main effect of each of nine network metrics (11 models), combinations of two uncorrelated network metrics (grooming or huddling with aggression) to examine their combined main effects (28 models), main effects plus a sex × dominance rank interaction (1 model), and main effects plus an interaction between each groom or huddle centrality metric and dominance certainty (7 models). From these, we selected our candidate model-set by selecting all models with a dAICc (or ΔAIC) < 2 (Burnham, Anderson & Huyvaert, 2011), and using the law of parsimony to eliminate models whose increased complexity does not improve AICc over a simpler model in the candidate set (Burnham, Anderson & Huyvaert, 2011; Grueber et al., 2011; Richards, 2005; Richards, 2008). We report coefficients and other summary statistics (P = 0.05 as ‘significant’, 0.5 < P < 0.1 as a ‘nonsignificant trend’) from each model within our candidate model-sets.

Results

Prevalence of Shigella flexneri in rhesus macaques

From the biochemical assays, we detected a moderate-low prevalence of Shigella among the CNPRC macaques. Specifically, Group I had the highest prevalence, with 23 out of 101 individuals being infected (∼23%). Groups II and III showed similarly low prevalence, i.e., 8 out of 96 (8.33%) and 7 out of 100 (7%) sampled individuals respectively.

Infection, network metrics, and individual attributes

In groups I and II, we found that well-connected individuals were socially buffered against the risk of Shigella infection. There were four models in the candidate model-set for the analysis of groups I and II (Table 1). These revealed significant, negative relationships between Shigella infection and three network metrics –groom out-degree (Model 4), groom eigenvector centrality (Model 7; Fig. 2), and huddle betweenness (Model 9) (Table 1). In other words, infection was least likely among individuals with strongest grooming and huddling connections. Models other than the candidate models also showed negative relationships between grooming and huddling connectedness and infection (e.g., groom betweenness: Model 5; huddle degree: Model 7; Table S2 ). Further, there was a significant interaction between huddling betweenness and dominance certainty (Model 20). When explored further, we found that individuals with categorically low certainties (below the 50th percentile) showed strong, negative relationships between huddling betweenness and infection (β =  − 4.19, p = 0.02) whereas those with categorically high dominance certainties showed no such effects (β = 0.38, p = 0.82). In other words, social buffering via huddling connections was more discernible among individuals with low dominance certainties than those with high dominance certainties. Finally, models that included aggression network metrics and/or sex interacting with dominance rank were not part of the candidate model set (Table 1; Tables S2 and S3), suggesting no clear relationships between these variables and infection.

Table 1 Summary of model statistics and parameter estimates from the candidate model set (all models with a ΔAIC < 2; outcome: Shigella presence–absence) for rhesus macaques in Groups I & II.

Model number	Predictor(s)	df	β	Adj. SE	P	AIC	AICc	Δ	w	
7	Intercept	195	0.83	0.39	0.035*	169.37	169.43	0	1	
	Groom eigenvector		−2.76	1.22	0.023*					
20	Intercept	193	−0.02	0.70	0.98	170.14	170.34	0.92	0.67	
	Huddle betweenness		−8.60	3.34	0.01*					
	Dominance certainty		−2.48	1.43	0.08a					
	Huddle betweenness : dominance certainty		13.55	6.06	0.026*					
4	Intercept	195	−0.91	0.41	0.026*	171	171.06	1.63	0.42	
	Groom outdegree		−2.31	1.16	0.047*					
9	Intercept	195	−1.14	0.32	0.001*	171.2	171.26	1.84	042	
	Huddle betweenness		−2.38	1.27	0.06a					
Notes.

* p < 0.05.

** p < 0.01.

a 0.05 < p < 0.1.

TITLEΔ Difference in AICc score from the best-fit model

w Relative model weight

Figure 2 Group I: grooming social network and Shigella infection risk.

Social Network graph of grooming relationships indicating the effect of social buffering on Shigella infection in Group I. Nodes are sized proportional to the eigenvector centralites of individual macaques. Infection (yellow nodes: N = 23 out of 101) is mostly restricted to individuals with the lowest eigenvector centralities.

In Group III, we found that Shigella infection was most prevalent among individuals that were involved in more aggressive interactions and/or more central in their huddling networks, suggesting contact-mediated transmission. Specifically, the candidate model-set constituted four models (Table 2) that revealed significant, positive relationships between Shigella infection and both aggression out-degree (Models 42, 11; Figs. 3 and 4) and out-strength (Models 41, 13). Models other than the candidate models also showed positive associations between possessing both direct and indirect grooming and huddling connections and Shigella infection (e.g., groom indegree: Model 4; huddle degree: Model 7; Huddle eigenvector: Model 9; Table S3). There was also a non-significant positive trend between infection risk and huddle betweenness (Models 41, 42) (Table 2 and Fig. 4). When we examined metrics from aggression networks with only intense aggressive interactions (bites, attacks, i.e., those involving contact), the positive relationships were sustained but nonsignificant (Model 42: contact-aggression out-degree: β = 3.37, p = 0.07; huddle betweenness: β = 2.87, p = 0.08). Finally, models that included grooming network metrics and/or sex interacting with dominance rank were not part of the candidate model set (Table 2; Tables S2 and S3), suggesting no clear relationships with infection.

Table 2 Summary of model statistics and parameter estimates from the candidate model set (all models with a ΔAIC < 2; outcome: Shigella presence–absence) for rhesus macaques in Group III.

Model number	Predictor(s)	df	β	Adj. SE	P	AIC	AICc	Δ	w	
44	Intercept	97	−5.47	1.33	0.001**	45.27	45.52	0	1	
	Aggression outstrength		5.09	1.81	0.005**					
	Huddle betweenness		3.42	1.87	0.067a					
42	Intercept	97	−5.97	1.52	0.001**	46.17	46.42	0.9	0.62	
	Aggression outdegree		5.12	1.95	0.008**					
	Huddle betweenness		3.31	1.84	0.072a					
13	Intercept	98	−3.93	0.73	0.001**	46.52	46.65	1.13	0.56	
	Aggression outstrength		4.56	1.61	0.004					
11	Intercept	98	−4.28	0.93	0.001**	47.30	47.43	1.91	0.38	
	Aggression outdegree		4.53	1.72	0.008**					
Notes.

* p < 0.05.

** p < 0.01.

a 0.05 < p < 0.1.

TITLEΔ Difference in AICc score from the best-fit model

w Relative model weight

Figure 3 Group III: aggression social network and Shigella infection risk.

Social Network graph of aggression relationships indicating the effect of social contact on Shigella infection in Group III. Nodes are sized proportional to the aggression out-degrees of individual macaques. Infection (yellow nodes: N = 7 out of 100) is prevalent among individuals with moderate-to-high out-degree.

Figure 4 Group III: aggression, huddling, and Shigella infection risk.

Scatter-plot showing the effect of social contact on Shigella flexneri infection for Group III. Black dots are infected macaques (N = 7 out of 100), and are concentrated among those with moderate-to-high huddle betweenness and aggression out-degree.

Discussion

In animal societies, the way in which sociality impacts individual health and fitness remains hotly debated. In regards to infectious disease susceptibility, increased social connections among group members may either facilitate the acquisition and transmission of pathogens via social contact (Drewe, 2010; Drewe & Perkins, 2015; Freeland, 1976; Loehle, 1995; MacIntosh et al., 2012), or may inhibit such acquisition via socially buffering individuals against daily stressors to reduce the risk of environmental acquisition of pathogens (Cohen et al., 2015; Hennessy, Kaiser & Sachser, 2009; Kaplan et al., 1991; Young et al., 2014). Results from our study speak to both of these processes. Specifically, they reveal that among two groups of rhesus macaques (a species characterized by strong social bonds Sade, 1972), having greater social connections in both grooming and huddling social networks socially buffered individuals against the risk of infection from an enteric pathogen, Shigella flexneri. Yet in a third group, we found that individuals who gave more aggression or functioned as centralized links or hubs of information flow between other individuals, were more likely to be infected, supporting the contact-mediated transmission hypothesis.

Nonhuman primates harbor a variety of pathogens, several of which also infect humans (Engel & Jones-Engel, 2011; Jones-Engel et al., 2005; Kaur & Singh, 2009; Nunn, 2012; Nunn & Altizer, 2006). In comparison with viruses and ectoparasites, the prevalence of zoonotic bacteria are relatively understudied (Nunn & Altizer, 2006). We found a moderate-to-low prevalence of Shigella flexneri in captive rhesus macaques, which is consistent with previous studies of bacterial prevalence in other free-living primate populations (e.g., free-living rhesus macaques: (Beisner et al., 2016), savannah baboons (Papio anubis: Drewe et al., 2012). Such comparisons reflect the general dearth of studies on bacterial prevalence among captive animals; on which studies to date have mostly focused on the clinical bases, i.e., the virulent versus asymptomatic effects, of infection among individuals (Lee, Kim & Park, 2011; Shipley et al., 2010). Indeed, the last documented reports of group- or population-specific prevalence levels of enteric bacteria among captive primates were on rhesus (23% of 4,476 individuals: Good, May & Kawatomari, 1969) and longtailed macaques (18.8% of 1,297 individuals: Takasaka et al., 1964) imported into biomedical facilities during the 60s. Given the now well-established, reliable characterization approaches (see ‘Methods’), our reports of low prevalence were not likely artifacts of methodological issues. Among humans on the other hand, reports of enteric bacterial prevalence are often inflated by sampling paradigms that focus purely on hospitalized patients, and/or individuals already showing signs of symptoms (e.g., Kotloff et al., 1999). We contend that pre-symptomatic, epidemiologically accurate assessments of enteric bacterial prevalence levels are imperative both for humans and captively housed animals given that these pathogens are (i) omnipresent, (ii) both socially and environmentally transmittable, and/or (iii) may cause unpredictable outbreaks of virulent infection both among humans and macaques (DuPont, 2000; Gupta et al., 2004; Kessler & Rawlins, 2016). Among the CNPRC macaques, a logical next step would be to establish associations between Shigella infection and symptomatic effects (e.g., diarrhea, enteritis) that may necessitate clinical treatment.

Animal studies that demonstrate the impact of social buffering on health and stress mitigation have traditionally focused on stress-related health outcomes in pair-housed or monogamous species (reviewed in Hennessy, Kaiser & Sachser, 2009), with evidence among socially cohesive, group-living species with strong social bonds emerging only more recently (e.g., Chacma baboons (Papio hamadryas ursinus): (Archie et al., 2014; Silk et al., 2010)); Barbary macaques (M. sylvanus: (Young et al., 2014); humans: (Holt-Lunstad, Smith & Layton, 2010)). Our findings extend the impact of social buffering to infectious agent acquisition. Among two groups of rhesus macaques, our candidate models revealed that individuals possessing direct and secondary grooming connections, and strong huddling connections particularly among individuals with uncertain dominance relationships, were the least prone to infection from Shigella flexneri. Broadly, these findings add novelty to epidemiological studies implementing social networks by contradicting the popularly prevailing notion that infectious agent acquisition is a consistent drawback of group living (Alexander, 1974; Freeland, 1976; Loehle, 1995; MacIntosh et al., 2012). The negative relationship between Shigella infection risk and each of grooming outdegree and eigenvector centrality points to grooming being a strong source of social buffering. This is consistent with prior research among primates that has revealed that both giving and receiving grooming may mitigate social stress (Henzi & Barrett, 1999; Smutt et al., 2007). Further, the establishment of the social buffering phenomenon in a second type of affiliation network (huddling) improves on most epidemiological studies that focus on just one (reviewed in Drewe & Perkins, 2015).

Specifically, the effect of grooming outdegree suggests that giving grooming may be more beneficial in lowering infection risk than receiving grooming, via potentially functioning to elicit additional buffering-related benefits such as social support (Seyfarth, 1977; Smutt et al., 2007). This argument is further supported by the effect of grooming eigenvector centrality, i.e., having a well-connected social circle of primary and secondary connections to elicit such support (Farine & Whitehead, 2015), on also lowering infection risk. Surprisingly, eigenvector centrality has been frequently overlooked in infectious disease research in favor of metrics based on direct connections (Drewe & Perkins, 2015). Yet one study (MacIntosh et al., 2012) found that grooming eigenvector centrality increased likelihood of infection from a nematode parasite (Strongyloides fuelleborni) in wild Japanese macaques, suggesting contact-mediated transmission. Our finding revealed an opposite, social buffering-mediated effect. Such an effect may be particularly discernible in study systems, such as large groups of captive macaques, that are socially (in addition to biologically: Suomi, 2011) analogous to dense human communities in urban settings wherein having a broad circle of social connections also has well-documented health- and fitness-related benefits (Janowski et al., 2012; Uchino, 2004; Uchino, 2009).

In addition to grooming connections, individuals with higher huddling betweenness were also more socially buffered against infection. Where social buffering prevails, betweenness, which measures the relative number of pathways that pass through each individual, may be a more reliable indicator of the strength and extent of support ties in networks (such as huddling) where the direction of the relationship (unlike in grooming) is less significant. Further, the effect of huddling betweenness in mitigating infection risk was particularly strong among individuals with more uncertain dominance relationships. Such a complex relationship between dominance status, network connectedness, and infection risk is consistent with what we see in the literature. The effect of dominance status on infection risk varies by social system and the type of health outcome examined. In primate groups for instance, high social rank is related to greater access to grooming partners and may enhance contact-associated susceptibility to infectious diseases (MacIntosh et al., 2012). Further, the relationship between dominance rank and stress (a proxy for infectious disease susceptibility via social buffering (Capitanio & Cole, 2015)) is highly inconsistent, with both high- and low-ranking individuals experiencing different types of social and biological stressors (e.g., Abbott et al., 2003; Sapolsky, 2005, but see Foerster et al., 2015). Thus dominance status, rather than linearly predicting infection risk, may instead interact with stress-levels and/or social network connectedness to influence this risk. To assess dominance status, we use the measure of certainty in addition to rank, which had no effects. Dominance certainty differs from rank; it is a metric of the predictability of the direction of an individual’s dominance interactions and pathways of interactions, irrespective of the outcome of wins or losses (Fujii et al., 2014, unpublished data). Biologically, it may be a more important moderator (than rank) on health outcomes. For instance, a recent study on the CNPRC rhesus macaques showed that individuals that face greater unpredictability in their dominance encounters also showed pronounced biomarkers of poor health, including inflammatory proteins and diarrhea (Vandeleest et al., 2016). Under such circumstances, the beneficial impact of possessing strong social connections in buffering individuals against infection risk may be more clearly discernible. A clearer picture on the effects of dominance on infection risk may emerge when future studies examine the effects of dominance certainty instead of or in addition to rank.

Our findings should lead to future investigations of whether increased network connectedness mitigates biological stress indicators (e.g., Glucocorticoid or GC content: Sapolsky, Romero & Munck, 2000; Young et al., 2014) and indeed, other systemic inflammation markers (e.g., C-reactive protein (CRP), Interleukin-6 (IL-6): Libby, Ridker & Maseri, 2002) that may influence buffering-mediated infection risk, which are currently under analysis. It is also conceivable that social connections may impact susceptibility versus resistance to bacterial infection via altering individuals’ foraging regimes and thereby modifying their gut microbial flora (Degnan et al., 2012; McCord et al., 2014). Among captive long-tailed macaques (M. fascicularis), for instance, commensal gut E. coli competitively inhibited infection from Shigella (Seekatz et al., 2013). Thus, assessing the links between social networks and stress- and/or microbiome-mediated infection risk would be logical next steps.

Social networks have proved to be highly beneficial in detecting contact-mediated infection and transmission of pathogens (Drewe & Perkins, 2015). Consistent with these previous efforts, we found that giving more aggression to others and having more huddling connections both increased individuals’ risk of Shigella infection in our third rhesus macaque group. Aggression may increase the risk of infection either via contact-risk (Drewe, 2010), or via weakening the health-related benefits of social buffering (Muller & Wrangham, 2004; Ostner, Heistermann & Schulke, 2008). The fact that aggression given, but not received, was related to infection risk better supports the former compared to the latter. Further, the positive association between huddling betweenness and infection risk suggests that individuals with increased huddling connections may be superspreaders of pathogens (Lloyd-Smith et al., 2005). This is because when contact-transmission is prevalent, betweenness, in its calculation of the relative number of pathways that pass through a node, specifically measures the potential for the flow of information (or pathogens) through a node (Drewe & Perkins, 2015; Farine & Whitehead, 2015; Newman, 2005; VanderWaal et al., 2014). It may therefore be an especially useful metric to parse out ‘informative’ nodes among captive groups where individuals tend to come into contact with more individuals than those in free-living groups, which are less spatially constrained (Drewe & Perkins, 2015; Griffin & Nunn, 2012). Finally, our findings yielded no information on Shigella transmission through Group III’s grooming network. This might be because grooming involves a more subtle form of hand-to-body-hair contact compared to huddling, which involves more direct and prolonged body-to-body contact that may increase the chances of transmission.

Although contact-mediated transmission in the aggression network might be expected to be most pronounced in a network composed of only intense contact-aggression, we found the reverse: a weaker association. One explanation for this could be that the detectability of transmission may be influenced by pathogen-specific differences in modes of transmission. For instance, the transmission of viral pathogens from macaques to humans would require the occurrence of specifically intense forms of contact, such as bites and scratches, which may culminate in the contact-exchange of body-fluids such as blood and saliva (Engel & Jones-Engel, 2011; Engel et al., 2013). Yet simpler body-contact and/or the sharing or consumption of contaminated food or water source has been shown to be sufficient for the transmission of enteric bacterial pathogens (like Shigella) among humans (Benjamin et al., 2013; Cooley et al., 2007).

In social systems, multiple additional factors may determine whether social buffering versus contact-mediated transmission of infectious agents may prevail. To our knowledge, most epidemiological studies that have reported contact-mediated transmission of infectious agents through social networks have focused on free-living or wild animal groups (reviewed in Drewe & Perkins, 2015). It is conceivable that social buffering may be more readily discernible among large, spatially constrained primate groups living in captivity. As in dense, suburban populations of humans (e.g., Janowski et al., 2012; Uchino, 2004; Uchino, 2009; see above), such living conditions manifest in more by-standing to witness (De Marco et al., 2010), and/or less opportunities for the social avoidance of, agonistic encounters (Fujii et al., 2014, unpublished data; McCowan et al., 2008). This may amplify the utility of strong social ties as avenues of support or stress-relief.

Yet by itself, living condition does not explain the heterogeneous pattern observed in Group III, which showed evidence for contact-mediated transmission. We offer two potential explanations for this. First, unlike Groups I and II which had diverse age-structures (3–22 year olds), Group III was primarily composed of younger individuals (3–11 years of age) (Table S1). Among free-ranging rhesus macaques, age proximity has been shown to positively influence the quality of affiliative social relationships (Widdig et al., 2001). Similarly, it may be likely that social ties among younger animals, on account of being more nascent and/or unpredictable, are not of the relationship quality that may be required for social buffering. Second, the discernibility of social buffering versus contact-transmission may be governed by higher-order social contexts, such as group stability. For instance, both Groups I and II maintained consistent dominance relationships with few reversals, as evidenced by consistencies in the overall directions of dominance encounters across their aggression and submissive status networks (Chan et al., 2013). In Group III, a comparison of these networks revealed marked inconsistencies in the direction of the relationships, which persisted until the group suffered a social collapse around 13 weeks after the data collection period (Chan et al., 2013). Thus it is conceivable that in captivity, contact-mediated transmission is more easily decipherable under socially unstable conditions where the effect of social buffering is minimal or absent. Validations of both these explanations await future assessments of (i) the age-classes of immediate connections, or “neighborhoods,” of older versus younger individuals, and (ii) expansions of current findings to additional groups.

In summary, this study suggests that within captive housed groups of a nonhuman primate biologically and socially analogous to humans, individuals’ network connections may socially buffer them against, or promote the contact-mediated transmission of infection from an enteric bacterial pathogen. The generality of these findings awaits expansions of similar approaches to additional social and pathogenic taxa. Such efforts may facilitate delineating what may be a fine line between when/how social network connections may be beneficial versus detrimental to infectious disease risk.

Supplemental Information

Figure S1 Diagnostic scatter-plots showing patterns of Shigella infection with respect to network metrics for individual macaques within each of Groups I, II, and III

In each plot, the Y-Axis represents the network metric plotted against network degree (a standard network metric) the X-axis. Black dots represent infected individuals (Group I: 23 of 100; Group II: 8 of 96; Group III: 7 of 99). Infection seems similarly concentrated among those with moderate-to-low degree and betweenness scores for Groups I and II, but more randomly scattered for Group III.

Click here for additional data file.

Table S1 Information on the age-structure, time of formation, and number of matrilines in each of the three study groups

Click here for additional data file.

Table S2 Complete model-set, model parameters, and summary statistics for all 48 models run for Groups I & II combined

Parameters in ‘Bold’ font represent those selected as candidate model-sets from each dataset, based on Δ < 2 and parsimony-based selection criteria.

Click here for additional data file.

Table S3 Complete model-set, model parameters, and summary statistics for all 48 models run for Groups III

Parameters in ‘Bold’ font represent those selected as candidate model-sets from each dataset, based on Δ < 2 and parsimony-based selection criteria.

Click here for additional data file.

Data S1 Dataset of Shigella presence–absence and network metrics for each of three groups of rhesus macaques analyzed in this study

Click here for additional data file.

We would like to thank our dedicated team from the McCowan Animal Behavior Laboratory, including A Nathman, A Barnard, T Boussina, A Vitale, E Cano, J Greco, N Sharpe, and S Seil, who participated in the behavioral data collection. We would also like to thank members of the Atwill and WIFSS (Western Institute for Food Safety and Security) laboratories, particularly C Bonilla, J Carabez, R Pisano, and I Wong, for playing key roles in the processing of fecal samples for pathogen isolation and characterization at the School of Veterinary Medicine, UC Davis. Finally, we are grateful to Dr. Hsieh Fushing and his research team at the Department of Statistics for the development and Percolation-and-Conductance method. We also thank him and his team along with Jian Jin for guidance in the implementation of the Percolation-and-Conductance (Perc) R package.

Additional Information and Declarations

Competing Interests

Author Contributions

Animal Ethics

Data Availability

The authors declare there are no competing interests.

Krishna Balasubramaniam conceived and designed the experiments, performed the experiments, analyzed the data, wrote the paper, prepared figures and/or tables.

Brianne Beisner analyzed the data, wrote the paper.

Jessica Vandeleest contributed reagents/materials/analysis tools, wrote the paper.

Edward Atwill conceived and designed the experiments, contributed reagents/materials/analysis tools, reviewed drafts of the paper.

Brenda McCowan analyzed the data, contributed reagents/materials/analysis tools, reviewed drafts of the paper.

The following information was supplied relating to ethical approvals (i.e., approving body and any reference numbers):

The protocols used for this research were approved by the UC Davis Institutional Animal Care and Use Committee were in accordance with the legal requirements of the jurisdictions in which the research was conducted.

The following information was supplied regarding data availability:

The raw data has been supplied as a Supplementary File and is available at Balasubramaniam, Krishna; Beisner, Brianne; Vandeleest, Jessica; Atwill, Rob; McCowan, Brenda. Data associated with Balasubramaniam, Beisner et al. (PeerJ, 2016): “Social buffering and contact transmission: Network connections have beneficial and detrimental effects on Shigella infection risk among captive rhesus macaques” (2016) DOI 10.15146/R3MK5W, http://n2t.net/ark:/c5146/r3mk5w.

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
