# Peer review of "Social buffering and contact transmission: network connections have beneficial and detrimental effects on Shigella infection risk among captive rhesus macaques"

_PeerJ, doi:10.7717/peerj.2630_

## Round 0.1 · original submission · Major Revisions

Thank you for your submission. Your article requires a number of major revisions which are listed in the reviewers comments below. You should address these comments prior to resubmission. In particular I would like to draw your attention to the following points;

- both reviewers noted concerns regarding how groups were assessed (reviewer 1 point 19 and reviewer 2 "validity of finding") and this should be addressed before resubmission.
- You should note both reviewers comments regarding referencing
- You should address the comments from reviewer 1 in the "validity of finding" section of their review (points 21-23)
I look forward to receiving your revised manuscript.

·

Basic reporting

I found this paper to be rather well-written, although a few points came to mind regarding the structure and content of the manuscript that could be improved:
1. L59 – Just a suggestion but it might not be a bad idea to diversify the set of citations referenced in this introductory paragraph. For example, while it’s not necessarily negative to cite a recent review paper discussing infection as a cost of social living, there are numerous original or concept-forming papers that have paved the way for this now common view, e.g. the seminal Alexander 1974 paper (already cited later in the present work) coming immediately to mind: “There is no automatic or universal benefit from group living… there are automatic and universal detriments… including… increased likelihood of disease and parasite transmission.” (p.328). Furthermore, on L59-62, aren’t there other papers of relevance here? For example, one can’t help to think about the numerous works by Sapolsky (also presented later in this ms) on stress and health.
2. L93-95 – it is a little misleading to use the term “therefore” here, because the previous lines were about grooming in nonhuman primates and before that about stress and non-communicable diseases in humans. It therefore doesn’t follow that “socio-positive interactions… decreases susceptibility to infectious agents”. Although this last point could be made as a standalone addition to the text.
3. L95-97 – I find it’s always better to hedge your bets here. If a statement like this must appear, perhaps a qualifier like “to our knowledge” is warranted?
4. L106-108 – I think it’s necessary to spell this out for the reader rather than force them to go to other publications to find information absolutely essential to understanding this paper. It should be clearly stated how shigella is transmitted and what we know e.g. about infective doses or whatever else we know that’s relevant to successful transmission of the organism, including factors related to the expression of disease in infected hosts. This becomes essential later when the SNA metrics are presented – i.e. what is the transmission mode of Shigella and are the social variables examined relevant?
5. L110 – this seems to be an abrupt transition from Shigella into the rhesus macaque model. One suggestion would be to separate this into its own paragraph, and then expand on the transmission pathways for shigella infection, and perhaps include more information on it’s known and/or predicted pathogenicity in its hosts (as suggested above).
6. L118-120 – I think this is a misleading statement. First, the citation should be Cohen et al 2015 (Psychological Science), not Killam 2015 in Scientific American, which is the popular version of the study. Second, the study can’t really discern between generally socially supportive (and supported) individuals and those that hug a lot, so how important the hugs themselves are remains to be seen. Third, the study did seem to show a buffering role for social support in that supported individuals had lower probability of becoming infected with the 2 experimental viruses, but there were no differences in clinical signs, i.e. expression of disease, between groups. “getting the flu” is really about the expression of disease, the flu-like symptoms, so it’s important to watch the wording here to avoid imprecision. Fourth, there is no evidence (at least presented) supporting a related function for huddling and hugging.
7. L125-127 – this may be generally or at least mostly true, but the paper by MacIntosh et al 2012 does include an analysis of the potential for physiological stress (via fecal glucocorticoids) to mediate infection, which it seemingly did not. Thus the study does attempt to discriminate between socially-mediated exposure v. susceptibility in that regard.
8. L127 – Also just a suggestion, but it’s a bit confusing to have the aims of the study spread out through the introduction in its current form. Perhaps laying out the background and then presenting the specific aims and predictions toward the end in a more traditional manner might help.
9. L236-237 – are you measuring infection here or pathogenic infection? Infection would just be the presence/absence of the infectious agent, but pathogenic infection would be an infection that has produced clinical signs of disease. The answer here is the former, but the authors need to be more careful in their descriptions of what is being studied. Similar change also required on L282. And in general, the terminology should be made consistent throughout, e.g. instead of saying “risk of pathogen infection” on L311, stick with Shigella infection – the authors admitted that little is known about pathogenesis in nonhuman primates.
10. L262-269 – First, I see little reason to present AIC values for best-fit models in the text, since AIC values can only be interpreted relative to other models in the candidate set. Second, why use “<” when describing p-values? Especially when values are over the traditional alpha level, but even when below, it is better to present exact statistics for precision. Third, this section seems to be presenting as ‘statistically significant’ results that are not statistically significant by traditional standards (alpha level @ 0.05). While I am not necessarily pro-Fisherian here, it should be made clear what criteria the authors are using to espouse statistical significance, particularly if using criteria that differ from the mainstream. Fourth, I now realize that the authors are presenting the results of the best-fit model, and then directing readers to the model averaged results presented in Tables 1&2. I see no reason to do so. As noted above, the practice of presenting results from models that ‘best fit’ a data set has come under strong scrutiny. The authors recognize that in using model averaging to lessen this potential danger. So why then present the results from ‘best fit’ models at all? These can be entirely misleading. And as noted above, without informing the reader how many models were in the candidate set, which sets of variables they contained, and their respective AIC values, it is very difficult to interpret the results of this study.
11. L269-272 – how were these effect sizes established for the different groups formed by rank stability? There doesn’t seem to be any mention of this in the methods.
12. L277 – why have an “e.g. out-degree” here? How was out-degree chosen as the representative of this group of analyses? This doesn’t seem logical. And in general, this list of results is a bit confusing. Perhaps it’s better to stick with presenting the model results in tabular format and simply refer to the table(s) in the text here to avoid such confusion and arbitrary reporting decisions.
13. L278 – here the authors were more cautious with the p<0.07 result: “somewhat more likely to...”. So, there is inconsistency within the results that needs to be corrected.
14. L317-322 – this section seems brief. Perhaps a discussion of the likelihood that the sampling method produced accurate results? Perhaps a further note about shigella infection in primates and other animals? What is the significance of infection, both in the wild and in captivity? How would captivity be expected to alter the results? Any or all of these make valid additions to the discussion.
15. L333-334 – the contrasting results for the different network metrics highlight the need to be clear about why each was included, what are the predicted relationships between each and infection, and why some were included and others not.
16. L353-354 – difficult to interpret the meaning of the Vandeleest study since it is currently in review. While I understand that the paper may be in a more advanced state of publication later, the authors may want to reconsider using it here without explanation of the main findings because it currently adds only confusion to the statement made here.
17. L356-360 – the authors might want to look at Foerster et al 2015 in Amer J Phys Anth for a study showing a negative relationship between rank and infection in guenons, also incorporating a measure of physiological stress.

Experimental design

In general, I found the experimental design sound. However, I do have a number of points to make regarding the reporting of the methods and statistics, and the statistical approach itself. My biggest concern is with grouping the two stable groups in the analysis, because it a prior reduces the chance of finding simple between group differences that could explain the results irrespective of the stability/instability argument.
18. L224 – what is the sensitivity of these shigella assays? Are rectal swabs the most reliable way to determine infection? Some discussion of this approach should be included to help readers assess the ability of this collection method to distinguish infected from uninfected individuals. Also, and importantly, since this is a managed population, to what extent are animals typically treated for this and other types of bacterial (and other) pathogens?
19. L231 – while I appreciate the rationale behind this, there may be a nontrivial problem with the way the groups were assigned here, namely that the results are assessed differently in groups 1 & 2 versus group 3. I’m not convinced that using ‘Group’ as a random effect can solve this problem, because overall you have 2 groups versus 1, a very small sample at the group level. Also, mathematically, random effects should have at least 5 levels for proper partitioning of variance, though to my knowledge it is still possible to use random effects for groups with fewer levels to limit pseudoreplication. Anyway, since we know that between group variability can be high, and for better comparison between groups here, I would recommend either replacing the current analysis with one for each of groups 1 & 2, or at the very least adding an analysis that fulfills this role. The authors might argue that inflating the number of models is also bad practice, but my thoughts are that the current comparison requires further evidential support that can only be gained via a true between group study design. Note that one alternative would be to create one model with all 3 groups included, and then use the interaction between tested variables and group stability, itself an added fixed effect, to determine directly whether there is a role for stability in this matter. That would provide a stronger test of the stability/instability hypothesis – currently the separate analyses only hint at this possibility but cannot discern it from simple between group variability.
20. L240 – I’m a bit confused on the approach. Here, the authors note a step-wise approach, but later they introduce the idea of candidate sets and AIC model selection and averaging. I think the latter approach is sound, but I wonder how the step-wise approach was implemented, because stepwise methods have been largely discredited in ecological research (e.g. among others Whittingham et al 2006, J Anim Ecol). Also, it’s impossible at present to get a sense of how many models were actually examined here. And related to that, how were the network metrics used chosen from among the long list of SNA metrics available? All of this information needs to be presented before the results can be interpreted properly.

Validity of the findings

The main conclusion of the study is that social networks influence infectious disease dynamics, but that social variables affect infection in different ways in stable versus unstable social groups. This is an intriguing result and may not be unexpected given the literature emerging on group stability and its various outcomes. However, I am not convinced that the analysis presented can distinguish a group stability effect from a simple between group variability effect. Stability was only one of the potential differences between these groups, and the one chosen by the authors. My first comment here relates mainly to this point, and is elaborated upon in the Experimental Design section above.
21. L313-316 – this seems to be the fundamental finding of this study, but following my comments to the analytical approach used, I would argue that this does not necessarily follow from the analyses and results presented. This claim could be made stronger if my suggestions above are incorporated. Otherwise, the authors should be much more cautious about making this claim, and present all possible alternatives.
22. L351 – this is debatable given that many of the results, as presented currently in the text at least, suggest only weak relationships between these metrics and infection (i.e. P {0.05-0.1})
23. L403 – I think it’s premature to talk about generalities based on this work alone, which investigates only shigella infection in captive groups of primates. And the study does nothing at all to address “general health and well-being” (L404-405). I think a lot more can be done to discuss the fact that this study was conducted in a captive setting, and how that might have impacted the findings. My aim in making this suggestion is not to deride captive studies, but to ensure that attention is paid to the fact that these animals are managed and cared for, and this must have some impact on disease dynamics in the population.

Additional comments

This paper presents and interesting and well thought out study of bacterial infection (Shigella) in captive rhesus macaques in relation to social networks and relative group stability, both of which are important social mediators of health outcomes. Particularly notable is the juxtaposition of socially-mediated exposure versus susceptibility, the comparison of stable and unstable social groups, and the multilevel analysis of group-level (stability) and individual-level (SNA metrics) social variables. Given that the expected relationships between the highly diverse social lives of primates and infectious disease dynamics are themselves numerous and diverse, this study presents a useful approach that should encourage further work in this area to take a broader perspective.

At the same time, I am not yet convinced that the conclusions are justified given the analyses and results presented. Above, I have offered numerous suggestions for the improvement of this manuscript, in the hopes that the authors can make their claims more convincingly.

·

Basic reporting

I find the manuscript „The benefits of buffering: social network connections inhibit pathogen transmission in stable groups of captive macaques“ by Balasubramaniam generally well and clearly written. The authors investigate the association of social connectedness with pathogen transmission vs. disease resistance via social buffering in three groups of captive macaques. In the dataset comprised of the first two groups, which are presented as “socially stable”, they find that macaques with more direct and also indirect connections in grooming and huddling networks have a decreased risk of being infected with Shigella flexneri, whereas in the third, “socially unstable” group animals with higher betweenness in the huddling network and a higher amount of aggression given were more likely to be infected. They conclude that overall social context determines whether social connectedness leads to increased disease transmission or decreased disease susceptibility via social buffering.
The research question is well laid out and put into context. However, the introduction can be improved by explaining that “social buffering” influences animal susceptibility / resilience to pathogens, whereas contact-mediated transmission is about increased exposure to pathogens. These are two distinct mechanisms! I suggest adding a sentence like “Social contacts can influence infection status in two ways, on the one hand they may lead to increased exposure to pathogens, on the other hand social contacts may decrease disease susceptibility via social buffering” at the beginning of the introduction. The same concerns the wording in the abstract and discussion sections:
In l. 35 I suggest using “contact mediated transmission” instead of just “social contact”, and in
l. 42/43 “more likely to be infected” rather than “more susceptible to infection”.
In l. 44 I suggest rephrasing to “…social connections may increase or decrease its chances of acquiring infection…”.
l. 54: “…strongly influence the risk of acquiring infectious disease”.
l. 63: “greater chances of contact-mediated transmission and…”
l. 83: “and decrease their susceptibility to infection”.

The authors have missed a few references concerning evidence for social buffering in wild primates, which I feel would strengthen their flow of arguments, e.g. after l. 95.
- Silk, J. B., J. C. Beehner, T. J. Bergman, C. Crockford, A. L. Engh, L. R. Moscovice, R. M. Wittig, R. M. Seyfarth and D. L. Cheney (2010). "Strong and Consistent Social Bonds Enhance the Longevity of Female Baboons." Current Biology 20(15): 1359-1361
- Archie, E. A., J. Tung, M. Clark, J. Altmann and S. C. Alberts (2014). "Social affiliation matters: both same-sex and opposite-sex relationships predict survival in wild female baboons." Proceedings. Biological sciences / The Royal Society 281(1793).

l. 99: “the risk of infection from a bacterial…”
l. 126: Again, I suggest using “contact mediated transmission” instead of just “social contact”.
l. 292: “thereby supporting contact-mediated transmission as the main effect of social contact.”
l. 304: “infectious disease risk”
l. 386: “supporting the hypothesis of increased disease transmission by social contact”.

Results section:
The fact that different statistical models were used and e.g. p-values differ between Tables 1 and 2 and the values given in the text is a bit confusing. Please state explicitly which values are derived from which models.

Experimental design

The research is original and in the scope of the journal. The research question is well defined and extremely relevant. The description of the methods is adequate. I recommend adding information on the time frame over which stability of groups I and II vs. group III was assessed, which is not explicitly stated (l. 228).

Validity of the findings

The authors interprete their statistical results in an appropriate manner, which is linked to the original research question. However, I am not fully convinced by the final conclusion given that only 3 social groups were compared, which may also differ in other aspects than social stability (e.g. sex ratio, age structure, environmental influences…). Therefore, the interpretation that “broader social contexts […] may delineate what seems to be a fine line between when/how social network connections may be beneficial […] vs. detrimental […]” needs to be toned down a little bit, emphasizing that this result still needs to be confirmed by other studies.
In the discussion section, I suggest discussing a few points in more detail:
- l. 317: The authors discuss prevalence of Shigella flexneri in just two sentences, comparing it to wild populations. This seems a bit meaningless, especially without any information on the pathogenicity of the organism (what kind of disease might it cause?) and the level of occurrence in the environment. Comparing wild and captive populations in terms of prevalence also does not make much sense as conditions regarding exposure and susceptibility may differ greatly between wild and captive populations. Comparisons with zoo populations / other captive populations in North america might be more meaningful. Otherwise the authors need to state why they think the comparison to wild populations is meaningful.
-The fact that Shigella flexneri can be transmitted via the environment and not only by social contact needs to be emphasized more.
- L. 326: Incorporate evidence for social buffering in wild primates, e.g. the two references provided above (Silk et al 2010, Archie et al. 2014).
l. 330: “grooming networks” instead of “groom networks”
l. 347 “…increased likelihood of infection… “
l. 349: Here, it may also be helpful to discuss that different components of the immune system are involved in the defense against bacterial pathogens vs. helminths, which may be differentially affected by stress or social buffering.

Additional comments

The research question is extremely relevant and this study is a great first step to initiate more research into how social connectedness impacts disease risk in the seemingly conflicting contexts of social buffering and contact-mediated transmission. I therefore highly recommend publishing this study after revisions concerning the aspects mentioned above have been carried out.

---

## Round 0.2 · Minor Revisions

Thank you for resubmitting your manuscript. It is significantly improved since the initial submission however the reviewer has raised a couple of minor editorial changes that should be address prior to any acceptance. In particular please note their comments regarding figure 1.

·

Basic reporting

The manuscript has been considerably improved since initial submission. Important background information and relevant citations have been included. I appreciate the change of the title and the toning down of interpretations regarding social stability. The discussion is now much more appropriate given the findings in the study.

Experimental design

I appreciate how the authors now describe their experimental design and that they offer more sound explanations of why the dataset was analyzed for groups I and II combined. However, I have some criticism regarding the diagnostic plots (Figure 1, see below).

Validity of the findings

I am still not convinced by the conclusion drawn from the analysis of the dataset from Group III (l. 347): The authors state that socially well-connected individuals have a higher infection risk in this group. What is “socially well-connected” in this context? In their set of candidate models, only aggression given and huddling betweenness are included as factors, thus, other metrics indicating social connectedness – and arguing much more for contact-mediated transmission – such as body contact (huddling/grooming) degree or strength, seem to have no effect? Please derive more specific conclusions, grounded on the data, here.

l. 372: “possessed more huddling connections” – This is not true, huddling degree was not included as a predictor in the candidate model set, only huddling betweenness was, which is a measure of indirect connectedness, not necessarily the number of connections. (also l. 475)

Additional comments

I find the use of different fonts in the text (bold and italic) somewhat irritating. No need to emphasize certain sentences that much – if the text is phrased well, readers will be able to grasp what is important without the use of different styles.
Figure 1 (diagnostic plots): I think the figure is not really appropriate for the data, as data points connected by a line suggest a paired-design / time series (e.g. before/after). Here, the data points represent different subsets of animals from the same group, however. I would also appreciate some information about the distribution of the individual data points around the means, either in the form of error bars, box-and-whisker plots or by showing all data points – but without the connecting lines between infected and uninfected subsets.
I also noticed a few points regrading grammar / style, which I annotated directly in the PDF.
Given that these minor revisions are carried out, I recommend publication of the manuscript.

---

## Round 0.3 · accepted · Accept

Thank you for completing the requested revisions. I am pleased to inform you that your manuscript has been accepted for publication.